# AMPNet: Asynchronous Model-Parallel Training for Dynamic Neural Networks

## Abstract

New types of compute hardware in development and entering the market hold the promise of revolutionizing deep learning in a manner as profound as GPUs. However, existing software frameworks and training algorithms for deep learning have yet to evolve to fully leverage the capability of the new wave of silicon. In particular, models that exploit structured input via complex and instance-dependent control flow are difficult to accelerate using existing algorithms and hardware that typically rely on minibatching. We present an *asynchronous model-parallel* (AMP) training algorithm that is specifically motivated by training on networks of interconnected devices. Through an implementation on multi-core CPUs, we show that AMP training converges to the same accuracy as conventional synchronous training algorithms in a similar number of epochs, but utilizes the available hardware more efficiently, even for small minibatch sizes, resulting in shorter overall training times. Our framework opens the door for scaling up a new class of deep learning models that cannot be efficiently trained today.

## 1 Introduction

An emerging category of neural networks show the common trait of reacting in dynamic and unique ways to properties of their input. Networks like tree-structured recursive neural networks (Socher et al., 2013; Tai et al., 2015) and graph neural networks (GNNs) (Scarselli et al., 2009; Li et al., 2015; Gilmer et al., 2017) take structured data types as input and and execute a computation that depends on the input structure. This defies the moden GPU-driven paradigm of minibatch-based processing, and we refer to this new class of models with dynamic control flow as *dynamic neural networks.*

The development of dynamic neural network frameworks – Chainer (Tokui et al., 2015), DyNet (Neubig et al., 2017a), and PyTorch (PyTorch core team) – speaks to the importance of this class of models and highlights the challenge of how to make it easy for users to describe them. Yet there is another big challenge: how can we *train* these models efficiently?

Managing minibatches to keep GPUs fully utilized is typically considered a user's responsibility in these dynamic frameworks (with the exception of DyNet's autobatching feature; see Sec. 7). This means that users have to think about how to change their data feeding pipeline or even the model itself to run efficiently on GPUs, rather spending time innovating to improve the model accuracy.

What if we had a hypothetical device with low memory overhead that allows perfect scaling without batching; i.e., processing 1 item is simply 100x faster than processing 100 items? Recent work on FPGAs and other specialized hardware (Farabet et al., 2011; Caulfield et al., 2016; Jouppi et al., 2017) for deep learning encourages us to investigate this question. Our premises are

1. No batching is required for efficient processing.
2. Each device may not have enough memory to hold the entire model (this is a realistic constraint for current memory systems that approach the perfect scaling we require)

Based on these premises, we propose an *asynchronous model-parallel* (AMP) training algorithm. Our idea is illustrated in Figure 1. We need model parallelism because each device may be too small to hold the entire model (premise 2). However, if we perform *synchronous* parameter updates following the full forward and backward propagations, the only way to increase device utilization is

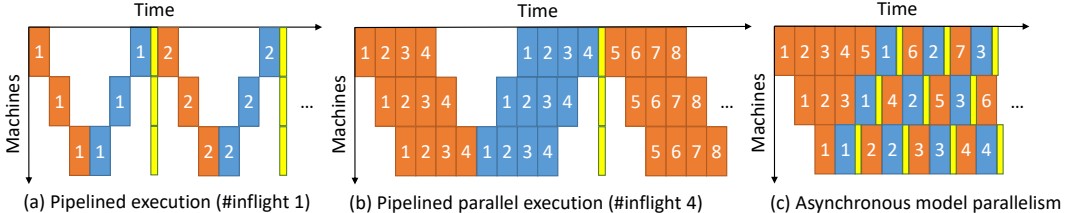

Figure 1: Gantt charts comparing pipelined synchronous model parallelism and asynchronous model parallelism. Orange, blue, and yellow boxes correspond to forward, backward, and parameter update operations, respectively. The numbers in the boxes indicate instance IDs.

by pipelining multiple instances into the system (see e.g., Chen et al., 2012). Pipeline parallelism with synchronous updates is at odds with convergence speed due to a decreased parameter update frequency; compare Fig. 1 (a) and (b).

To overcome this problem, we propose *asynchronous* parameter updates that occur without global synchronization whenever a pre-specified number of gradients have been accumulated; see Fig. 1 (c). With this design we aim for both high device utilization and update frequency.

In this setting, however, model parameters may be updated between the forward and the backward computation of an instance, introducing gradient "staleness". Despite staleness, we show that AMP training can converge fast with good hardware utilization. Specifically, our contributions are:

- We present the asynchronous model parallel training algorithm for efficient distributed training of dynamic networks.

- We present an intermediate representation (IR) with explicit constructs for branching and joining control flow that supports AMP training. Unlike previous work that considers static computation graphs for static control flow (e.g., Caffe), and dynamic computation graphs for dynamic control flow (e.g., Chainer), our IR encodes a static computation graph to execute dynamic control flow[1]. This makes training easy to distribute and parallelize.

- We show that our IR can readily encode *replicas*, a form of *data parallelism* (see Sec. 5). In addition, our IR includes operators for data aggregation, which recover a form of *batching*, enabling our methods to be applied even on hardware where batching is beneficial.

- We implement AMP training on a multi-core CPU and empirically demonstrate that AMP training converges to similar accuracies as synchronous algorithms on a variety of dynamic neural network models including Tree RNN and gated graph neural networks (GGNN).

In summary, our work demonstrates the benefits of AMP training and gives a novel way to design and deploy neural network libraries with dynamic control flow. In addition, we use our implementation to estimate the performance on a hypothetical device satisfying premises 1 & 2, with 1TFLOPS compute capability (see Appendix C). Together, these contributions open up new ways to scale up dynamic networks on interconnected compute devices.

## 2 NEURAL NETWORKS WITH DYNAMIC CONTROL FLOW

Below we highlight three models with dynamic control flow that will be studied in depth in this paper:

*Variable-length RNNs* iterate over the tokens of variable-length sequences. Pseudo-code for a simple vanilla RNN is given in Figure 2. The linear (fully connected) layer and rectified linear unit (ReLU) can be substituted with a more sophisticated unit such as a gated recurrent unit (Chung et al., 2014). Though each instance has a different length, it *is* possible to add padding to enable batching. However this may lead to significant redundant compute due to variability in sequence lengths.

---

[1]Our IR bears similarity to TensorFlow but we discuss differences in Section 7.

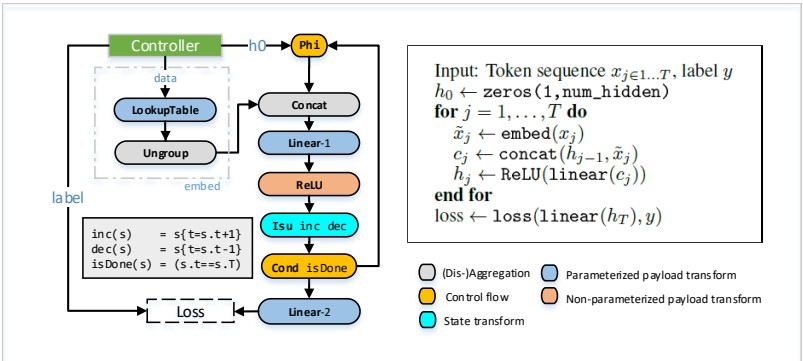

Figure 2: Variable-length RNN in IR and pseudocode (colors denote IR node types)

---

**Algorithm 1** Asynchronous Model-Parallel training

---

**while** training not done **do**
    **for all** workers in parallel **do**
        Wait until a message arrives
        msg ←the highest priority message from the input queue
        op ← sink node of the message msg
        **if** msg is *forward* type **then**
            Compute forward: out_msgs ← op(msg)
            Enqueue the resulting message(s) into the queue(s) of the workers hosting the child nodes.
        **else if** msg is *backward* type **then**
            Compute backward: out_msgs ← op*(msg, fwd_msg)
            Enqueue the resulting message(s) into the queue(s) of the workers hosting the parent nodes.
        **else if** msg is *update* type **then**
            Perform parameter updates on op
        **end if**
    **end for**
**end while**

---

*Tree-structured neural networks* are powerful models used for parsing of natural language and images, semantic representation, and sentiment analysis (Socher et al., 2011; Bowman et al., 2016; Socher et al., 2013; Tai et al., 2015). They require evaluation of (potentially multiple) trees with shared parameters but different topology for each instance. Each tree structure is instance-specific and batching requires nontrivial planning (Looks et al., 2017). A simple form of tree neural network performs a bottom up traversal of the instance, starting from an embedding of the leaves. At each level the values from the child nodes are concatenated and sent through a specialized unit (e.g. LSTM). The result is then propagated further up the tree. Backpropagation over the tree structure is known as backpropagation through structure (Goller and Kuchler, 1996).

*Graph neural networks* (Scarselli et al., 2009; Li et al., 2015; Gilmer et al., 2017) combine both the temporal recurrence and recurrence over the structure. GNNs can be seen as performing aggregation/distribution operations over a general graph structure with shared parameters.

Apart from the models above, there exist many recently proposed models with flexible control flow (e.g. hierarchical memory networks (Chandar et al., 2016), neural programmer interpreters (Reed and De Freitas, 2015), adaptive computation networks (Graves, 2016; Figurnov et al., 2016), and sparsely-gated mixture of experts (Shazeer et al., 2017), to which our framework can be applied.

## 3 ASYNCHRONOUS MODEL-PARALLEL TRAINING

In AMP training, each node of a computation graph (including control flow nodes – see next section) is associated with a *worker*, which is an abstraction of a compute device. Neural network training and inference is carried out by message passing among workers following algorithm 1. All workers run in parallel without synchronization. Each message contains a *payload* (float or int tensor), as

well as a *state* that includes the IDs of the source and sink nodes and a label indicating the type of message (forward, backward, or update). The state is used to keep track of algorithm and control flow information. For example, in a variable-length RNN the state also contains the instance identifier, the current position in the sequence, and the total sequence length for the instance. More generally, if the neural model use (possibly nested) loops, then the state for the messages that arrive to and are produced from nodes that logically belong in loop bodies will contain sets of loop counters that together with the instance id uniquely identify the messages throughout the course of the computation.

When a worker receives a message labeled as *forward* (or *backward*) then it performs the operation of the node indicated by the sink node ID on the supplied payload. This produces one or more outgoing messages that are then enqueued into the queues of the workers hosting the next sink nodes in the computation graph. The final loss layer initiates backward propagation. If the message type is *update*, the worker will carry out weight updates on the sink node using gradients accumulated in the appropriate slot in the worker's local memory. Since both the operation weights and weight gradients can be stored locally on the worker then workers only need to communicate activations and activation gradients, which are typically an order of magnitude smaller than the weights. The update message is typically sent from the sink node to itself as part of the backward process but it can also be sent from a *controller* node to simulate synchronous pipelined training.

There are two important details that are not fully spelled out in Algorithm 1. First, since the messages arrive asynchronously (and possibly out of order), any operation that has more than one parent nodes need to store the payload into its local cache until all the parents send the corresponding payloads. Thus output message(s) can only be produced when all the payloads become available. The cache needs to be able to distinguish payloads received from different parents and payloads with different instance ID, and different counters (all encoded in the message states). The same is true in the backward pass for a node with multiple child nodes. Second, $\text{op}^*$ denotes the adjoint operation of $\text{op}$ and takes the backward message $\text{msg}$ and potentially the forward message $\text{fwd\_msg}$ stored in the cache. For a nonlinear activation node (e.g., ReLU), the node will not change the state of the message in the forward pass. Thus in the backward pass, the adjoint operation will just multiply the partial derivative of the activation function to the payload of the received backward message keeping the state unchanged. By contrast, an operation that only changes the state of the message in the forward pass (e.g., increment the loop counter) will reverse the change in the backward pass leaving the payload unchanged.

In the experiments we vary two hyper parameters to control the effect of asynchrony:

`min_update_interval`: determines the minimum number of gradients that a parameterized operation needs to accumulate before it can update its parameters (using the *update* message). The staleness of a gradient can be measured by the number of updates between the forward and backward computation that produces the gradient. Small `min_update_interval` may increase gradient staleness. On the other hand, large `min_update_interval` can reduce the variance of the gradient but can result in very infrequent updates and also slow down convergence.

`max_active_keys`: controls the maximum number of *active instances* that are in-flight at any point in time. By setting `max_active_keys = 1` we restrict to single-instance processing, typically equivalent to synchronous training. More in-flight messages generally increase hardware utilization, but may also increase gradient staleness. Section 6 demonstrates the effects of these parameters in a multi-core CPU runtime.

## 3.1 Worker Affinity

In our asynchronous execution model, the optimal assignment of $N$ neural network computation graph nodes to $W$ workers (referred to as *affinitization*) is in general a non-trivial scheduling problem. We investigated several heuristics for assigning affinities (for example k-hop coloring (Ichrak et al., 2012) to ensure subsequent heavy operations were assigned different workers). However, we find that the following procedure achieves high performance in practice in our multi-core CPU implementation, and is adopted throughout our experiments for simplicity. We first partition the nodes into $H$ 'heavy' operations (namely matrix multiplies) and $(N - H)$ 'light' operations, and then balance the heavy nodes across the workers by affinitizing the $h^{\text{th}}$ with the $(h \mod W)^{\text{th}}$ worker. Finally, the light operations are affinitized randomly among the rest of the workers. Note that in scenarios

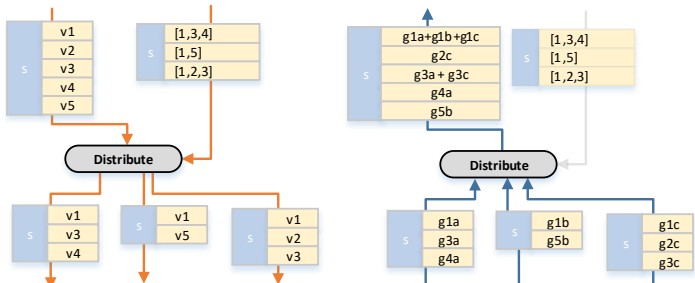

Figure 3: Distribution based on dynamic information

where communication is over a physical network affinitization will become more critical for high performance.

## 4 A STATIC INTERMEDIATE REPRESENTATION FOR DYNAMIC CONTROL FLOW

**Overview** Computation graphs are expressed using a static intermediate representation (IR) that can be a compilation target for high-level libraries (e.g. TensorFlow or our own Python and C++ frontends), and can itself admit multiple backends (e.g. the multi-core CPU runtime in this paper, or a network of accelerators). Static means that the IR graph is *instance-independent*. Nevertheless, it can execute dynamic and instance-dependent control flow decisions, in a forward and backward manner, by storing instance- and iteration-dependent information as the computation evolves. Each IR node comes with a forward and a backward semantics. A model is specified by (i) an IR graph, and (ii) a specialized controller loop that pumps instances and other data (e.g. initial hidden states or labels), and is responsible for throttling asynchrony. In the rest of this section we discuss the most important IR nodes along with their operational semantics, and show how they are used in the example models.

**Payload transformations** *Parameterized payload transform* (PPT) nodes can be used to encode, for instance, fully connected layers. They apply a transform in the forward pass, but also record the activation in order to use it to compute gradients in the backward pass. An activation is recorded by keying on the state of the message, allowing thus to process forward and backwards messages completely asynchronously, and – in the extreme case – out of order. A PPT node requires specification of the forward and the backward transformation. It may decide to independently apply accumulated gradients to update its parameters. For transformations that do not involve parameters (e.g. ReLUs) our IR offers a simpler *non-parameterized payload transform*.

**Loops, state, and control flow** A *condition* node (Cond $f$) is parameterized by a function $f$ that queries the *state* (but not the payload) of the incoming message and, based on the response, routes the input to one of the successor nodes. A *join* node (Phi) propagates the messages it receives from each of its ancestor nodes but records the origin (using the state of the message as the key) so that in the backward pass it can backpropagate them to the correct origin. An *invertible state update* node (Isu $f$ $f^{-1}$) is parameterized by two functions $f$ and $f^{-1}$ that operate on the state of a message, and satisfy $f^{-1}(f(x)) = x$. Typically these are loop counter update functions.

Figure 2 shows how to encode an RNN. The loop at the heart of the RNN is implemented with Cond, Phi and Isu nodes. The controller pumps sequences in a lookup table (just another PPT layer), and our Ungroup node (to be described in the next section) generates a stream of tensors each corresponding to a single token, tagged with the current time-step (loop counter). For each forward message, the Isu node increments the time-step, and the conditional node tests whether the end of the sequence has been reached. Depending on the answer it either propagates the hidden state back to Phi, or pushes the hidden state to the final linear and loss layers. In backward mode, messages pass through the Isu (which decrements the time-step), and reach the Phi node. The Phi node will (based on information from the forward phase) either back-propagate to the Cond node, or to the controller to terminate. Hence the loop is executed in both the forward and backward direction.

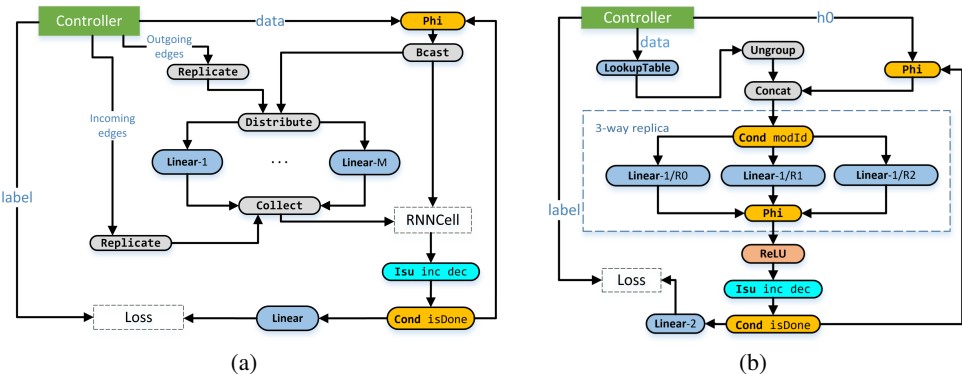

Figure 4: IR graphs for (a) Gated Graph Neural Network and (b) RNN-with-replicas. The RNNCell in (a) denotes a recurrent structure (e.g. GRU, LSTM), the details of which we omit. `Cond modId` in (b) implements a round-robin branching to different replicas by computing (instance ID mod 3).

**Aggregation and disaggregation**   Our IR offers several constructs for aggregation and disaggregation; for example RNN requires us to concatenate (`Concat`) hidden states and embeddings with matching timesteps and instance ids (states). We offer a construct for broadcasting (`Bcast`) a message to multiple successors. We offer an ungrouping construct (`Ungroup`) that ungroups a matrix and emits all resulting messages tagged with an extra user-provided increasing loop counter. This allows us to insert the stream of token embeddings in the middle of the RNN loop in Figure 2. In backward mode `Ungroup` groups back all the incoming gradients. A simpler variant of `Ungroup` is `Replicate`, which replicates a message with an extra loop counter in the state. In backwards mode `Replicate` sums up all incoming gradients that correspond to the state without the extra loop counter.

Figure 4(a) describes a GNN that combines aggregation on the structure of a graph instance with an outer loop. The controller pumps data that contain the feature embeddings of all nodes of an input graph. In addition it pumps in a map specifying graph topology. The `Distribute` node uses that information along with the graph itself to create sub-matrices (here each corresponding to edge types) and pass them through linear (fully connected) layers. The `Collect` node collects and re-groups the results based on graph topology. These nodes correspond to a form of dynamic partition and merging. Schematically the `Distribute` behaviour is given in Figure 3. In backward mode, based on the control information received during the forward pass (gray line) re-groups the gradients and sums together those that correspond to the same index. The `Collect` operator is essentially symmetric.

## 5   DATA PARALLELISM IN MODEL PARALLELISM

Pipelined model parallelism can often be augmented with forms of data parallelism. Consider the RNN in Fig. 2. The only heavy operation (Linear-1) in the body of the loop will act as a bottleneck for computation. One solution is to split the linear layer into smaller tiles and compute them in parallel. This is expressible in our IR but the linear operation needs to be large enough to benefit from tiling in this way. Another approach is to replicate the linear layer in full. This requires only minimal new machinery – we can replicate the linear layer and place the replicas inside `Cond` and `Phi` nodes as in Figure 4(b). Different instances or messages from the same instance but with different position in the sequence can be processed in an (pipeline-)parallel fashion by being sent to one of the replicas chosen by a random or deterministic function of the message state. To enable parameters to be shared among the replicas, we use infrequent end-of-epoch replica synchronization (averaging) that incurs negligible communication cost. We also tried more elaborate message-passing protocols for group synchronization, but found that infrequent global synchronization was sufficient for fast convergence.

## 6   EXPERIMENTS

We evaluate AMPNet using the dynamic models introduced in Section 2. For completeness, we additionally consider a multi-layer perceptron (MLP) as an example of a static, batched network

that AMPNet is not specifically designed to tackle. Brief details of the models and data sets in the experiments are presented below, and further details are given in Appendix B.

| Model | Data set | Difficulty | Experiment purpose |
|---|---|---|---|
| MLP | MNIST | toy | *The* standard deep learning benchmark (for which AMPNet is not specifically designed) |
| RNN | List reduction | toy | The simplest architecture with dynamic control flow (a `for` loop) |
| Tree-LSTM | Stanford Sentiment Treebank | real-world | The TensorFlow Fold benchmark for backpropagation through structure |
| GNN | QM9 | real-world | Complex control flow with loops over time and structure |

**Results**   Our asynchronous runtime is motivated by the promise of emerging hardware (e.g. FPGA accelerators) that fulfill the premises in section 1 and are well suited to dynamic neural network execution. Here we are primarily interested in how the performance of our runtime improves as we increase the degree of asynchrony (by varying `max_active_keys`) while keeping other factors fixed. The aim is to answer the question of whether AMP training is a promising direction for novel distributed hardware that deviates from the CPU/GPU batching paradigm. To answer this question using resources available today we run the AMPNet training using a multi-core CPU runtime where each worker is a hardware thread (see Appendix A). Additionally we forecast the performance on a hypothetical 1TFLOPS device satisfying our premises by replacing all computation nodes by configurable sleep nodes. This allows us to estimate the performance on a new hardware keeping the control flow decisions dynamic. See Appendix C.

It is also interesting to compare how the raw CPU performance of the AMP runtime compares with existing frameworks (TensorFlow, TensorFlow Fold and DyNet) to see that our implementation is already competitive with state of the art methods even on CPUs that do not conform to our target hardware model. We provide additional analysis for each experiment below.

On MNIST, Table 1 shows 3x speedup from synchrony (`max_active_keys = 1`) to asynchrony (`max_active_keys = 4`) in terms of throughput. This is almost ideal as the first three linear layers are the heaviest operations. The number of epochs to reach the target validation accuracy increases from 3 to 4 but the overall speedup in terms of the wall clock time is 2.2x. We have also compared AMP training against pipeline parallel training (Fig. 1 (b)). Figure 5(a) shows that while AMP training achieves 3x throughput gain already with `max_active_keys = 4`, pipeline parallelism can only achieve 2x (in fact from Fig. 1 (b), $3m/(m+2)$ is the best case for `max_active_keys = m`)

Table 1: Time to convergence to target validation accuracy. The time to convergence can be broken down into number of epochs and the throughput (instances/s). The target accuracy is shown inside parentheses next to each dataset. `mak` is a short-hand for `max_active_keys` defined in Sec. 3; `mak = 1` corresponds to synchronous training for MNIST and minimal asynchrony arising from just one in-flight instance for other models with recursive structures.

| | AMP | | | | TensorFlow | | |
|---|---|---|---|---|---|---|---|
| | `mak` | time (s) | epochs | inst/s | time (s) | epochs | inst/s |
| MNIST (97%) | 1 | 91.1 | 3 | 2076 | **34.5** | 3 | 5880 |
| | 4 | 40.4 (2.2x) | 4 | 6268 | | | |
| List reduction (97%) | 1 | 78 | 10 | 13k | 46 | 7 | 18k |
| | 4 | 66 (1.2x) | 10.5 | 16k | | | |
| | 16 | 60 (1.3x) | 9.5 | 16k | | | |
| (2 replicas) | 4 | 38 (2x) | 12 | 35k | | | |
| (4 replicas) | 8 | **21 (3.8x)** | 13 | 71k | | | |
| Sentiment (82%) | 1 | 185 | 2 | 102 | 208 | 5 | 265 |
| | 8 | **82 (2.3x)** | 2 | 222 | | | |
| | 16 | 104 (1.8x) | 3 | 258 | | | |
| QM9 (4.6) | 4 | N/A | N/A | 988 | 6696 | 66 | 1217 |
| | 16 | **5227** | 76.5 | 1797 | | | |

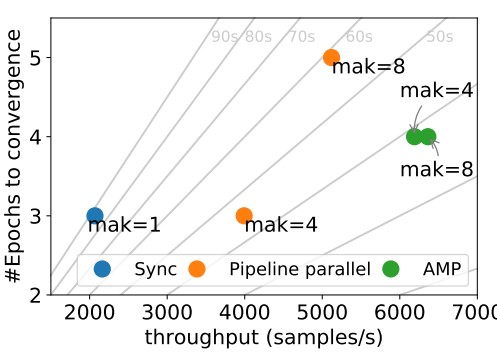 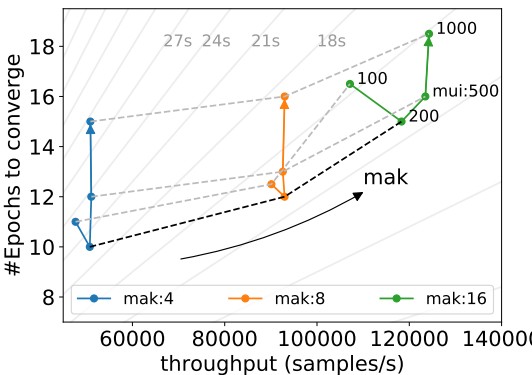

(a) Comparison against pipeline parallelism (Fig. 1 (b)) for 4-layer MLP on MNIST.

(b) Performance of an 8-replica RNN model on the list reduction dataset as a function of asynchrony hyperparameters.

Figure 5: Throughput-convergence trade-off as a function of mode of parallelism and asynchrony hyper-parameters. Solid gray lines show constant convergence time trajectories; mak and mui stand for `max_active_keys` and `min_update_interval`, respectively.

and higher throughput at `max_active_keys` = 8 is achieved at the cost of the sharp increase in the number of epochs to convergence.

The list reduction dataset demonstrates the power of *replicas*. As there is only one heavy operation (Linear-1, Figure 2), the speedup from asynchrony is mild (1.3x). However we get 2.5x and 3.5x speedup for 2 and 4 replicas, respectively, which is nearly ideal. Again, the # of epochs to convergence is not affected by increasing `max_active_keys`. The slowdown in convergence for 4 replicas is due to the increased *effective* minibatch size – also commonly observed in data parallel training.

Next the sentiment tree-RNN dataset shows that our runtime is competitive without batching to TensorFlow Fold (Looks et al., 2017) using dynamic batching of batch size 100. It is worth mentioning that our runtime allows us to specify different `min_update_interval` parameter for each parameterized operation. We set this parameter to 1000 for the embedding layer, which is initialized by Glove vectors, and 50 for all other layers. This reduced gradient staleness in the embedding layer.

The QM9 dataset demonstrates that increasing asynchrony helps on real-world tasks with complex control flow, and our method outperforms an efficient TensorFlow implementation on CPUs.

Finally, we have implemented the BiLSTM w/ char model in Neubig et al. (2017b) on Wikiner dataset Nothman et al. (2013). Our preliminary implementation without any batching achieves 130 sentences/s at `max_active_keys` = 32 without any noticeable loss in accuracy (around 94 % after one epoch). This is competitive with DyNet's performance on the same machine (23 sentences/s without and 220 with autobatching, respectively); See Sec. B.5 for details.

**Asynchrony** Finally we provide additional analysis on the effect of asynchrony. The degree of asynchrony is controlled by hyperparameters `min_update_interval` and `max_active_keys`. In Fig. 5(b) we use an 8-replica RNN model on the list reduction dataset to investigate how these parameters affect the data and time required to converge to 96% validation accuracy. We find, in analogy with minibatch size in traditional systems, that `min_update_interval` must neither be too large nor too small. Increasing `max_active_keys` (increasing asynchrony) monotonically increases performance when the number of keys is similar to the number of individually affinitized heavy operations in the model 8 in this case). Increasing `max_active_keys` significantly beyond this point produces diminishing returns.

## 7  RELATED WORK

One approach to the task of training networks with instance dependent control flow is to define the computation graph dynamically per-instance. This approach is taken in Chainer (Tokui et al.,

2015), DyNet (Neubig et al., 2017a), and PyTorch (PyTorch core team). There are key challenges in accelerating this approach: Model parallelism would require the dynamically generated computation graph to be scheduled on-the-fly, and BLAS level parallelism would require operations to be batched on-the-fly. Automatic dynamic batching has been implemented in DyNet (Neubig et al., 2017a), and is an interesting alternative to our asynchronous execution. Similar methods are used in TensorFlow Fold (Looks et al., 2017). The basic idea is to inspect and merge together (by depth) the unrolled computation graphs of several instances to create batched BLAS operations. The effectiveness of automatic batching greatly depends on the model – for example, it would not perform well on random permutations of a sequence of operations. By contrast, our IR would very succinctly express and achieve pipeline parallelism using a static computation graph that is easy to distribute and optimize.

Theano (Al-Rfou et al., 2016) and TensorFlow (TF)(Abadi et al., 2016) can syntactically handle instance dependent control flow with abstractions for conditional execution (`ifelse` in Theano and `cond` in TF) and loops (`scan` and `while_loop`, respectively); TF also provides higher-order functions, such as `map`, `foldl`, `foldr`, and `scan`. The main difference between AMPNet and the above frameworks is that AMPNet is streaming and asynchronous whereas Theano is non-streaming and synchronous. Although not designed for streaming, TF can support streaming programmatically as it exposes first-class queues, as well as data prefetching with so called *input pipelines*. In our IR, all the queuing is implicit and stream-based execution is the default. TF additionally does support static description of dynamic control flow and state update, but we depart from the classic dataflow architecture that TF follows (Arvind and Culler, 1986): First, instead of having nodes that represent mutable reference cells, we encapsulate the state with which a message should be processed through the graph in the message itself. Second, because we encapsulate algorithmic state in the messages, we do not introduce the notion of control dependencies (which can be used to impose a specific execution order on TF operations). Our choices complicate algorithmic state management from a programming point of view and make the task of designing a high-level compiler non-trivial, but allow every node to run asynchronously and independently without a scheduler and without the need for control messages: For example, nodes that dynamically take a control flow path or split the data simply consult the state of the incoming message, instead of having to accept additional control inputs. For "small" states (e.g. nested loop counters or edge and node ids) this might be preferable than out-of-band signaling. Our IR can implement loops by simply using state-update, conditional, and phi nodes, because the state accompanies the payload throughout its lifetime, whereas TF introduces specialized operators from timely dataflow (Murray et al., 2016) to achieve the same effect.

Asynchronous *data* parallel training (Recht et al., 2011; Dean et al., 2012; Chilimbi et al., 2014) is another popular approach to scale out optimization by removing synchronization (orthogonal to and combinable with model-parallel training). For example, convolutional layers are more amenable to data-parallel training than fully connected layers, because the weights are smaller than the activations. Moreover, when control flow differs per data instance, data parallelism is one way to get an effective minibatch size $> 1$, which may improve convergence by reducing variance. The impact of staleness on convergence (Recht et al., 2011) and optimization dynamics (Mitliagkas et al., 2016) have been studied for data parallelism. It would be interesting to extend those results to our setting.

Jaderberg et al. (2016), like us, aim to to train different parts of a model in a decoupled or asynchronous manner. More precisely, their goal is to approximate a gradient with a *synthetic gradient* computed by a small neural network that is locally attached to each layer. Hence, the local gradient calculation becomes independent of other layers (except for the training of the gradient predictor network) and allows asynchronous parameter updates. This would be especially useful if the evaluation of the local network is cheaper than the computation of the real gradient; for example, if the computation of the real gradient required significant communication of forward/backward messages between devices.

## 8 CONCLUSION AND OUTLOOK

We have presented an asynchronous model-parallel SGD algorithm for distributed neural network training. We have described an IR and multi-core CPU runtime for models with irregular and/or instance-dependent control flow. Looking forward, we aim to deploy our system on specialized hardware. Equally importantly, we plan to build a compiler that automatically deduces the information to be placed in the states and generates state keying functions from a higher-level description of the models. By unlocking scalable distributed training of dynamic models, we hope to enable exploration

of this class of models that are currently only on the horizon but may become more mainstream in the future.

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

## A    AMPNET RUNTIME IMPLEMENTATION

We have implemented an AMPNet runtime for multi-core CPUs. Our runtime spawns multiple *workers* each associated with a hardware thread and hosting one or more IR nodes – in a more general setting each worker corresponds to a compute device. To remain faithful to a distributed environment communication is only through message passing. Each worker is equipped with a multiple-producer single-consumer queue that can accept messages for any IR node hosted on that worker.

The main worker loop periodically offloads messages from the concurrent queue to a worker-local priority queue that assigns higher priority to backward messages. Backward prioritization is designed for situations when multiple IR nodes with a dependency on the IR graph end up hosted on the same worker. As a consequence, backpropagation can complete faster and new instances can be pumped in by the controller. We dequeue the top message and invoke the forward or backward method of the target IR node. These methods may update internal IR node state (such as cache the state of the incoming message and wait for more messages) or post new forward or backward messages.

How to update the parameters using the gradients is a configuration option that selects amongst a range of optimization algorithms. We have implemented runtime configuration options for selecting several well-known schemes such as (momentum-)SGD and Adam Kingma and Ba (2014), and for controlling the training hyper-parameters.

## B    DETAILS OF THE EXPERIMENTAL RESULTS

We provide more details of the experiment and analysis in this section. All experiments were carried out on machines with 16 cores and 112 GB of RAM. The validation curves were averaged over at least 20 independent runs. The time/epoch to reach a target accuracy was calculated as *median* of the time an algorithm takes to reach the target accuracy over the repetitions. We found this approach to be more reliable than reporting the time/epoch when the averaged accuracy reaches the target. Table 2 shows both the training and validation throughputs we obtained with AMPNet and our TensorFlow baselines.

### B.1    MNIST

We train a 4-layer perceptron with ReLUs and 784-dimensional hidden units on MNIST (LeCun and Cortes, 1998). Both the AMP runtime and a TensorFlow baseline use SGD with learning rate 0.1 and batch size of 100.

Figure 6(a) shows the validation accuracy vs. time, validation accuracy vs. epochs, and throughputs of synchronous and asynchronous versions of AMPNet as well as TensorFlow. The throughput greatly increases from synchronous (max_active_keys = 1) to asynchronous (max_active_keys = 4) while the speed of convergence (middle panel) is hardly affected for mild amount of asynchrony. Taking higher max_active_keys = 8 increase throughput only very little (because there is no more work) and seems to rather make the convergence more unstable. This is due to the fact that our current scheduler is greedy and pumps in a forward message whenever the first layer is unoccupied, which leads to large gradient staleness. Clearly a better scheduling will remove this sensitivity.

### B.2    LIST REDUCTION DATASET

We train a vanilla RNN to perform reduction operations on variable length lists of digits. Each training instance is a sequence of at most 10 tokens: The first token indicates which of 4 reduction operations [2] is to be performed, and the remaining tokens represent the list of digits. The output is the result of the calculation rounded modulo 10. The dataset consists of $10^5$ training and $10^4$ validation instances.

We present this task as a classification problem to a vanilla RNN with ReLU activation and a hidden dimension of 128. All parameterized operations are affinitized on individual workers. We bucket training instances into batches of 100 sequences (in the baseline and in AMPNet).

---

[2] The operations considered in our toy dataset act on a list $L$ and are expressed in python syntax as: `mean(L)`, `mean(L[0::2])-mean(L[1::2])`, `max(L)-min(L)` and `len(L)`.

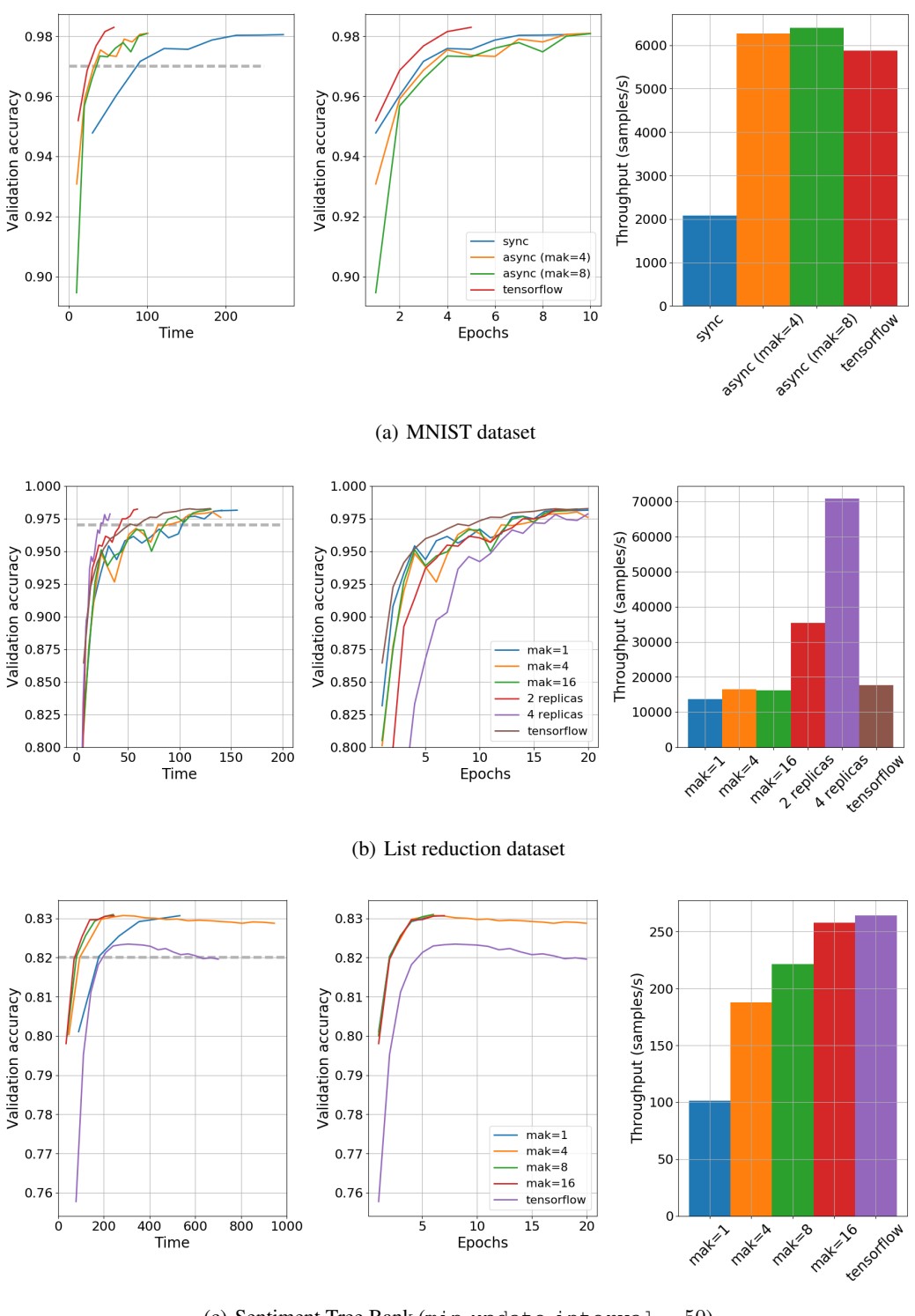

(a) MNIST dataset

(b) List reduction dataset

(c) Sentiment Tree Bank ($\mathtt{min\_update\_interval} = 50$)

Figure 6(b) shows the validation accuracy vs. time and the number of epochs, and throughputs of the methods we discussed in the main text on the list reduction dataset. We first notice that increasing the asynchrony from synchronous ($\mathtt{max\_active\_keys}$=1) to $\mathtt{max\_active\_keys} = 4$ and $\mathtt{max\_active\_keys} = 16$ affects the convergence very little at least in average. However, there

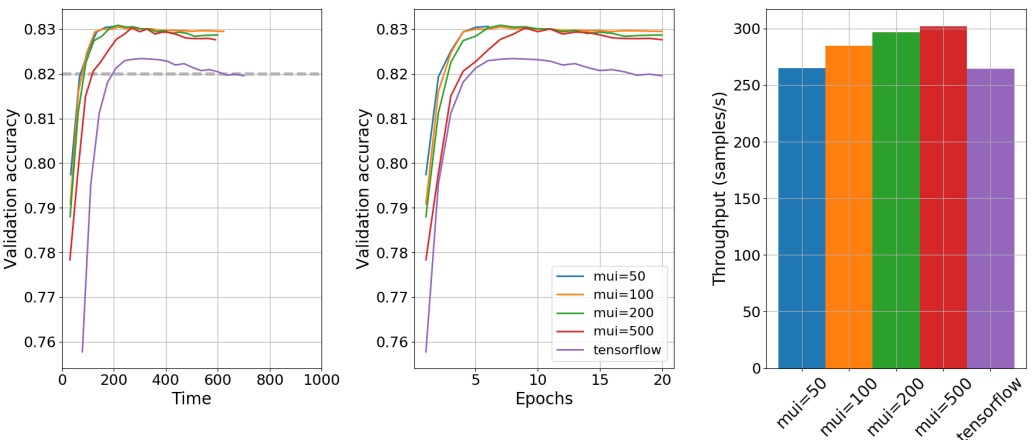

(d) Sentiment Tree Bank (`max_active_keys = 16`)

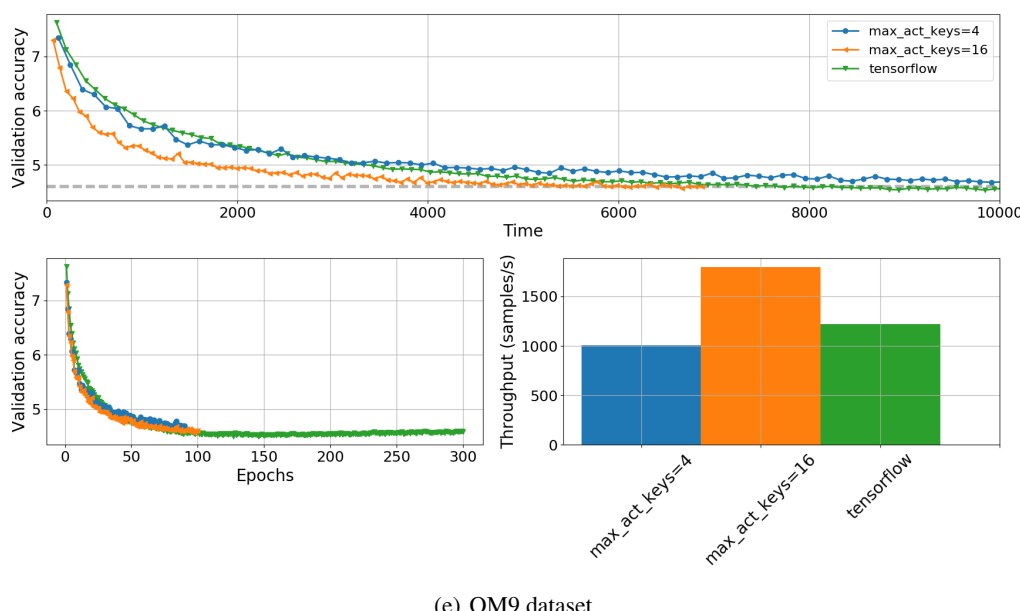

(e) QM9 dataset

is also very little speedup without introducing replicas as we discussed in the main text. Increasing the number of replicas increases the throughput almost linearly from 13k sequences/s (synchronous) to 35k sequences/s (2 replicas) and over 70k sequences/s (4 replicas). Convergence is almost unaffected for 2 replicas. This was rather surprising because the parameters of the replicas are only synchronized after each epoch as we described in Sec. 5. A slight slow-down in convergence can be noticed for 4 replicas. Since even `max_active_keys = 16` has almost no effect on the convergence without replicas, this is not due to asynchrony. We also tried to synchronize more frequently but this did not help. Thus we believe that the slow-down is due to the increase in the effective minibatch size resulting in reduced number of updates per epoch, which is commonly observed in data parallel training.

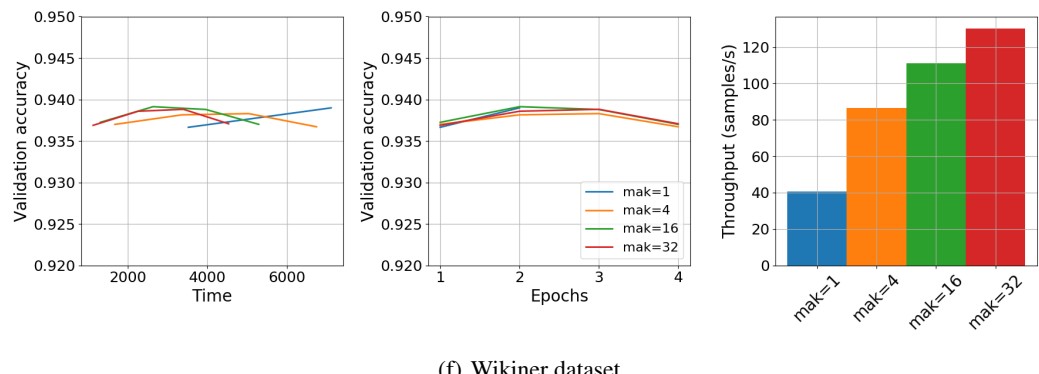

(f) Wikiner dataset

Figure 6: Convergence plots.

Table 2: Training and validation throughputs.

| | number of instances | | AMP | | | TensorFlow | |
| --- | --- | --- | --- | --- | --- | --- | --- |
| | train | valid | `mak` | train inst/s | valid inst/s | train inst/s | valid inst/s |
| MNIST (97%) | 60k | 10k | 1 | 2076 | 6094 | 5880 | 8037 |
| | | | 4 | 6268 | 18973 | | |
| List reduction (97%) | 100k | 10k | 1 | 13k | 39k | 18k | 43k |
| | | | 4 | 16k | 55k | | |
| | | | 16 | 16k | 56k | | |
| (2 replicas) | | | 4 | 35k | 100k | | |
| (4 replicas) | | | 8 | 71k | 178k | | |
| Sentiment (82%) | 8511 | 1101 | 1 | 102 | 280 | 265 | 1583 |
| | | | 4 | 188 | 692 | | |
| | | | 16 | 258 | 822 | | |
| QM9 (4.6) | 117k | 13k | 1 | 302 | 710 | 1217 | 3218 |
| | | | 4 | 1007 | 2585 | | |
| | | | 16 | 1797 | 4845 | | |
| Wikiner (94 %) | 142k | 1.7k | 1 | 41 | 101 | – | – |
| | | | 4 | 86 | 284 | | |
| | | | 32 | 130 | 426 | | |

## B.3 SENTIMENT TREE BANK DATASET

We consider the sentiment classification dataset from Socher et al. (2013) consisting of binarized constituency parse trees of English sentences with sentiment labels at each node. Following Tai et al. Tai et al. (2015), we use 8,544 trees for training, 1,101 trees for validation, and 2,210 trees for testing.

We use a Tree LSTM for this classification task based on the TensorFlow Fold Looks et al. (2017) benchmark model. Both the AMP and Fold models are trained following Tai et al. (2015) with the additional architectural modifications proposed by Looks et al. (2017); Semeniuta et al. (2016). Furthermore, we split our Tree-LSTM cell into Leaf LSTM and Branch LSTM cells. This does not affect the expressiveness of the model because the LSTM cell receives either zero input (on branch) or zero hidden states (on leaves); i.e., the two cells do not share weights except for the bias parameters, which are learned independently in our implementation. We compare the time to reach 82 % fine grained (5 classes) accuracy (averaged over all the nodes) on the validation set.

Figure 6(c) shows the averaged fine grained validation accuracy for the tree RNN model with different `max_active_keys` on the Stanford Sentiment Tree Bank dataset. Interestingly although TensorFlow Fold achieves higher throughput, AMPNet converges faster (in terms of the number of epochs). This speedup is mainly due to the fact that we are not batching and updating whenever we have accumulated 50 gradients (except for the lookup table node that updates every 1000 gradients); 50 gradients correspond to roughly 2 trees. The reason for the lower throughput compared to

TensorFlow Fold is that we are only grouping the leaf operations and not the branch operations. Grouping the branch operations is possible by extending our IR nodes and we are actively working on it.

Figure 6(d) shows the same information for fixed `max_active_keys` = 16 and different `min_update_interval`. We can see that as we increase `min_update_interval` from the originally used 50 to larger values, the peak of the validation accuracy shifts later and lower becoming closer to the curve obtained by TensorFlow Fold. This is consistent with the parallels between `min_update_interval` and minibatch size we drew in Section 6. The `min_update_interval` parameter has marginal influence on the training throughput.

### B.4 QM9 DATASET

We study a real-world application for GNNs: prediction of organic molecule properties from structural formulae in the QM9 dataset Ruddigkeit et al. (2012); Ramakrishnan et al. (2014). GNNs have previously been applied to this task in Gilmer et al. (2017).

We concentrate on prediction of the norm of a molecule's dipole moment using a regression layer build on the propagation model from Li et al. (2015) (corresponding to the simplest setting in Gilmer et al. (2017)). We use a hidden dimension of 100 and 4 propagation steps, initializing the graph nodes (atoms) following Gilmer et al. (2017). The molecules contain up to 29 atoms and in a TensorFlow baseline we bucket molecules into batches of 20 with atom counts differing by at most 1 within a batch. Following Gilmer et al. (2017), we report regression accuracies in multiples of a target accuracy from the chemistry community.

Figure 6(e) shows that GGNN can tolerate relatively large `max_active_keys` = 16, and increased the throughput significantly from 300 graphs/s (synchronous) to 1797 graphs/s (see Table 2).

### B.5 WIKINER DATASET

We compare the performance of AMP training with DyNet with and without autobatching using the BiLSTM tagger w/ char model from the DyNet benchmark suite[3]. This model consists of both character-level and word-level bidirectional LSTMs. The model uses a learnable word embedding for frequent words (more than 5 times in the corpus) and character-level bidirectional LSTMs for infrequent words. We use our `Distribute` and `Collect` nodes to dynamically route the messages depending on the word frequencies.

Wikiner dataset[4] is a named entity recognition dataset extracted from Wikipedia by Nothman et al. (2013). We use the training/validation split provided by the DyNet benchmark suite with 142,153 and 1,696 sentences, respectively.

The result is shown in Fig. 6(f). We achieve more than 3x in throughput from `max_active_keys` = 1 to `max_active_keys` = 32 without any noticeable loss in the validation accuracy after 2 or 3 epochs. The slight decrease in the validation accuracy after the third epoch is due to overfitting and it is not related to asynchrony.

### C THROUGHPUT ESTIMATION FOR A HYPOTHETICAL 1 TFLOPS DEVICE

In order to estimate the performance of AMPNet on a hypothetical device with 1 TFLOPS compute capability, we replaced all fully connected layers in the network with a dummy operation that simply waits for a specified time. The dummy operation waits for $2 \cdot k \cdot d_{\text{in}} \cdot d_{\text{out}} \cdot 10^{-12}$ seconds when the input is $k \times d_{\text{in}}$ and the weight matrix is $d_{\text{in}} \times d_{\text{out}}$ for forward, backward, and gradient accumulation operation. It waits for $d_{\text{in}} \cdot d_{\text{out}} \cdot 10^{-12}$ seconds for weight update. In this way we can maintain all the data-dependent control decisions (e.g., sequence length) identical to the original network and also measure the real time spent for all the other operations. In order to calculate the time to reach target accuracy, we take the median number of epochs the original network required to reach the target

---

[3]`https://github.com/neulab/dynet-benchmark/`
[4]`http://schwa.org/projects/resources/wiki/Wikiner`

accuracy and calculate the time as

$$\mathtt{time} = \mathtt{epochs} \cdot \left( \frac{n_{\mathrm{train}}}{\mathtt{throughput}_{\mathrm{train}}} + \frac{n_{\mathrm{val}}}{\mathtt{throughput}_{\mathrm{val}}} \right),$$

where $n_{\mathrm{train}}$ and $n_{\mathrm{val}}$ are the number of training and validation instances, respectively, and $\mathtt{throughput}_{\mathrm{train}}$ and $\mathtt{throughput}_{\mathrm{val}}$ are the throughput for training and validation, respectively.

The results are shown in Table 3. For the 4-way replicated RNN, we estimate roughly 260k instances/s, which is a 3.7x speedup compared to our CPU runtime. For tree RNN and GGSNN, we estimate milder 30 –70 % speedups mainly due to the fact that they have more complicated operations like `Distribute` and `Collect`, which we did not attempt to extrapolate the computation time because the implementation on a new hardware may be drastically different from the current CPU runtime.

Table 3: Estimated time to convergence to target validation accuracy and training throughput with a hypothetical 1 TFLOPS device. `mak` is a short-hand for `max_active_keys` defined in Sec. 3; $\mathtt{mak} = 1$ corresponds to synchronous training for MNIST and minimal asynchrony arising from just one in-flight instance for other models with recursive structures.

| | | AMP | | | | 1 TFLOPS (estimated) | |
|---|---|---|---|---|---|---|---|
| | target accuracy | mak | time (s) | epochs | inst/s | time (s) | inst/s |
| List reduction | 97% | 1 | 78 | 10 | 13k | | |
| (4 replicas) | | 8 | 21 | 13 | 71k | 5.15 | **268k** |
| Sentiment | 82% | 1 | 185 | 2 | 102 | | |
| | | 16 | 104 | 3 | 258 | 60.6 | **440** |
| QM9 | 4.6 | 16 | 5227 | 76.5 | 1797 | 3827 | **2455** |

