# OpenReview forum: "AMPNet: Asynchronous Model-Parallel Training for Dynamic Neural Networks"
_ICLR.cc/2018/Conference — Reject_

### Official Review · AnonReviewer3 · 2017-11-25
**It is an interesting paper about model parallelism for dynamic neural networks. The proposed methodology is novel, but the implementation is not very complete.**

**Rating:** 6
**Confidence:** 4

**Review:**

The paper describes a model-parallel training framework/algorithm that is specialized for new devises including FPGA. Because of the small memory of those devices, model-parallel training is necessary. Most current other frameworks are for model parallelism, so in this sense, the framework proposed by the authors is different and original. The framework includes a few interesting ideas including using intermediate representation (IR) to express static computation graph and execute it as dynamic control flow, combining pipeline model parallelism and data parallelism by splitting or replicating certain layers, and enabling asynchronous training, etc.

Some concerns/questions are
1) The framework is targeted at devices like FPGA, but the implementation is a multicore CPU SMP. It makes the computational result less convincing. Also, does the implementation use threading or message passing?
2) Pipeline model parallelism seems need a lot of load balance tuning. The reported speedup results confirm this conjecture. Can the limitation of pipeline model parallelism be improved?

Page 4, in the "min_update_interval" paragraph, why "Small min update interval may increase gradient staleness."? I would think it decreases staleness.

The paper is clearly written and easy to follow.

---

> ### Author Response · Authors · 2017-12-06
> **reply**
>
> Thank you for your comments. Please find our response to your questions below
>
> "Also, does the implementation use threading or message passing?"
> the implementation is based on C++ threads, and the communication between threads is all done by explicit message passing in order to remain faithful to a truly distributed execution.
>
> "Page 4, in the "min_update_interval" paragraph, why "Small min update interval may increase gradient staleness."? I would think it decreases staleness. "
> This is because when min_update_interval is small, the updates occur more frequently, so the parameters may change several times between the forward and backward passes of a data instance. Consider the limiting case of 1 update per epoch (i.e. maximum min_update_interval) - in this limit there is no gradient staleness because all the forward and backward messages see the same parameters. Reducing min_update_interval from this limit will monotonically increase gradient staleness. Increasing staleness is not necessarily a problem for the convergence rate, and Fig 5. shows that tuning min_update_interval can lead to good performance.

---

### Official Review · AnonReviewer1 · 2017-11-27
**AMPNet review**

**Rating:** 4
**Confidence:** 5

**Review:**

This paper presents AMPNet, that addresses parallel training for dynamic networks. This is accomplished by building a static graph like IR that can serve as a target for compilation for high-level libraries such as tensor flow. In the IR each node of the computation graph is a parallel worker, and synchronization occurs when a sufficient number of gradients have been accumulated. The IR uses constructs such as concat, split, broadcast,.. allowing dynamic, instance dependent control flow decisions. The primary improvement in training performance is from reducing synchronization costs.

Comments for the author:

The paper proposes a solution to an important problem of model parallel training especially over dynamic batching that is increasingly important as we see more complex models where batching is not straightforward. The proposed solution can be effective. However, this is not really evident from the evaluation. Furthermore, the paper can be a little dense read for the ICLR audience. I have the following additional concerns:

1) The paper stresses new hardware throughout the paper. The paper also alludes to “simulator" of a 1 TFLOPs FPGA in the conclusion. However, your entire evaluation is over CPU. The said simulator is a bunch of sleep() calls (unless some details are skipped). I would encourage the authors to remove these references since these new devices have very different hardware behavior. For example, on a real constrained device, you may not enjoy a large L2 cache which you are benefitting from by doing an entire evaluation over CPUs. Likewise, the vector instruction processing behavior is also very different since these devices have limited power budgets and may not be able to support AVX style instructions. Unless an actual simulator like GEM5 is used, a correct representation of what hardware environment is being used is necessary before making claims that this is ideal for emerging hardware.

2) To continue on the hardware front and the evaluation, I feel for this paper to be accepted or appreciated, a simulated hardware is not necessary. Personally, I found the evaluation with simulated sleep functions more confusing than helpful. An appropriate evaluation for this paper can be just benefits over CPU or GPUs, For example, you have a 7 TFLOPS device (e.g. a GPU or a CPU). Existing algorithms extract X TFLOPs of processing power and using your IR/system one gets Y effective TFLOPs and Y>X. This is all that is required. Currently, looking at your evaluation riddled with hypothetical hardware, it is unclear to me if this is helpful for existing hardware. For example, in Table 1, are Tensorflow numbers only provided over the 1 TFLOPs device (they correspond to the 1 TFLOPs column for all workloads except for MNIST)?  Do you use the parallelism at all in your Tensorflow baseline?  Please clarify.

3) How do you compare for dynamic batching with dynamic IR platforms like pytorch? Furthermore, more details about how dynamic batching is happening in benchmarks mentioned in Table 1 will be nice to have. Finally, an emphasis on the novel contributions of the paper will also be appreciated.

4) Finally, the evaluation appears to be sensitive to the two hyper-parameters introduced. Are they dataset specific? I feel tuning them would be rather cumbersome for every model given how sensitive they are (Figure 5).

---

> ### Author Response · Authors · 2017-12-06
> **reply**
>
> Thank you for your careful review.
>
> First, there seems to be a potential misunderstanding. In your summary you write "In the IR each node of the computation graph is a parallel worker, and synchronization occurs when a sufficient number of gradients have been accumulated" but we would like to point out that there is no need for and we do not perform synchronization because each operation is pinned to its worker (in our model parallel paradigm, there is only one copy of each operation's parameters, and they are stored in memory directly accessible from the worker hosting the operation) . The only exception is where we combine data parallelism with model parallelism using replicas: Here, replicas are synchronized at the end of each epoch.
>
> Please find our response to your questions below:
>
> 1) The hypothetical computational capability of 1 TFLOPS is configurable depending on the hardware characteristics. We feel that many new devices are candidates for neural network accelerators. While it will eventually be important to understand fine details of a chosen hardware's characteristics, it is not our intention to lay out detailed calculations for one specific hardware. Instead, we want to point out a model class and training algorithm where our CPU based simulator and back-of-the-envelope calculations show that real value could be added by hardware that significantly differs from batch-based processors like GPUs.
>
>
> 2.1) We would like to stress that only comparing the effective flops (which is equivalent to comparing only the throughput for a given model) can be misleading because we can always maximize the device utilization by sacrificing the rate of convergence. For example, increasing the minibatch size for GPU based training increases device utilization, but spending time processing large batches to produce a small number of low variance gradient steps can lead to overall slower convergence than taking many (higher variance) gradient steps with small minibatches. This is why we focus on the time to reach target validation accuracy in our experiments and report both the throughput and the number of epochs to convergence.
>
> 2.2) Sorry for the confusing vertical alignment of TensorFlow in Table 1. TensorFlow uses the same number of threads and is run on the same hardware as our multi-core implementation. We will separate out any references to the 1 TFLOPS device in our revised version.
>
> 3) Dynamic frameworks such as PyTorch make it easy for users to write dynamic models but there is no advantage in terms of efficiency, and in particular, it is difficult to deploy dynamic models written in PyTorch in an distributed environment. In contrast, our contribution is to express dynamic neural networks using a static intermediate representation which is critical for training these models in a distributed environment.
>
> 4) In Figure 5 we have elucidated how the two parameters influence the speed of convergence in terms of throughput and convergence rate. In practice, tuning these parameters is not too difficult: max_active_keys can be set to the number of CPU cores, and min_update_interval has a similar interpretation to the batch size which is common in other neural network frameworks.

---

> > ### Comment · AnonReviewer1 · 2017-12-11
> > **response**
> >
> > How do you enforce min_update_interval in the no data parallel case? Is there a wait or any other synchronization primitive?
> >
> > 1) For this paper to be accepted, it is necessary that the FPGA claims and the use of the word 'simulator' be removed. A conference paper is not the place to make strong claims using back of the envelope calculations. There are other papers, at least in other communities that are using programmable FPGA boards or real simulators using Gem5 to build model/data parallel software systems. Hence, I feel the claims need proper evaluation or needs to be removed. As I mentioned in my review, if this works fine on existing parallel CPUs or GPUs, it is sufficient for the paper to be accepted.
> >
> > 2) If you find effective TFLOPs to be misleading, which is extensively used to compare performance improvement for many convolution algorithms and other recent papers in ICLR. Please provide any metric that shows that you actually improve performance over *existing model parallel systems*.  I find it rather strange that a paper about speeding up deep learning computations uses sleep() functions to slow down execution in its evaluation to establish a baseline.
> >
> > Your response "effective TFLOPs (..) can be misleading because we can always maximize the device utilization by sacrificing the rate of convergence" weakens the motivation of the paper as shown in Figure 1. Your response implies that Figure 1(a) can converge faster than Figure 1(c), and Figure 1(c) may not even be desirable.
> >
> > "Dynamic frameworks such as PyTorch make it easy for users to write dynamic models but there is no advantage in terms of efficiency, and in particular, it is difficult to deploy dynamic models written in PyTorch in an distributed environment. In contrast, our contribution is to express dynamic neural networks using a static intermediate representation which is critical for training these models in a distributed environment."
> >
> > 3) I am surprised that you make such strong statements about deploying pytorch while providing no evaluation results to backup your claims.  Even though it appears that you are convinced that your paper will work better on all FPGA devices in the future and is  superior to pytorch/Dynet in every aspect, a reader needs  a scientifically correct and thorough evaluation to be convinced. More importantly, a clear evaluation tells us when this works and when it doesn't.
> >
> > 4) Please provide more intuition on "min_update_interval has a similar interpretation to the batch size"? So a very small min_update_interval i.e synchronous has better convergence? Is this dataset specific or independent of the problem? Unlike batch sizes which are driven by GPU sizes and memory limits and are rather well known, I get the intuition that this hyper parameter will require multiple runs of experimentation for each dataset and it may just be faster to run a synchronous version.
> >
> > I have updated my score based on your response. I am looking forward to a revised version and I will update my score based on how you fix the evaluation and claims as pointed out in 1, 2 & 3.

---

> > > ### Author Response · Authors · 2017-12-20
> > > **more clarification**
> > >
> > > 1) We will move the extrapolation to a 1 TFLOPS device and all other quantitative analysis of potential performance on new hardware to an appendix following the reviewer's suggestion. Our main contribution is to propose and empirically verify the working of asynchronous model parallelism as an alternative to mini-batch-based data parallelism. Although we think it is interesting to analyze how this algorithm might perform on distributed systems of new devices, we are happy to move these quantitative claims to an appendix to make the contributions of the main text clearer.
> > >
> > > 2) The metric used in Table 1 to demonstrate the benefits of our method is wall clock time to reach a pre-specified validation accuracy on identical hardware. We believe that this is the most relevant metric because raw TFLOPs or throughput comparison only makes sense when two approaches achieve the target validation accuracy the same number of epochs (see e.g. Goyal et al. [1]; see also [2, p30]). Note that we also show the number of epochs required to converge in Table 1.
> > >
> > > To achieve a good wall clock time, the algorithm needs to do two things: (1) process instances quickly (high throughput) and (2) have a high parameter update frequency. Fig 1a and Fig 1b are intended to show two extremes: Fig 1a has low throughput and high update frequency; Fig 1b has higher throughput but lower update frequency. Fig 1c shows that asynchronous model parallelism could achieve the best of both worlds, and it is the purpose of this paper to demonstrate that there is promise for this training algorithm.
> > >
> > > Note that we *do* compare synchronous model parallelism (corresponding to Figure 1a) in our experiments (see max_active_keys=1). We can see consistent speed-up in the time to reach target accuracy which can be explained by increased throughput and only a slight increase in the number of epochs to convergence. Figure 5 explores this trade-off in more detail.
> > >
> > > 3) We never claimed  that "our paper will work better on all FPGA devices in the future and is superior to pytorch/Dynet in every aspect". Neural network deployment in a distributed environment is a challenge that we certainly need to do more work (currently we run on multiple threads). The only point we made was the dynamic construction of computational graph (define-by-run principle) employed in dynamic frameworks does not make it easier to achieve this goal because any graph optimization/partition/scheduling that can be done dynamically can be implemented when the computational graph is statically available but not the other way around.
> > >
> > > 4) min_update_interval is the number of gradients that a parametrized operation computes and aggregates before applying an update to its parameters. A small min_update_interval leads to frequent, noisy updates, and a larger value gives infrequent but less noisy updates. Note that this is analogous to batch size in batch based training. In batch based training, the optimal batch size needs tuning and is intimately linked with learning rate tuning (less noise in gradient estimates allows increased learning rates). Similarly, some tuning is required with min_update_interval (see Fig. 5).
> > >
> > > [Clarification about the need for synchronization]
> > > We would like to point out again that we do not require any synchronization for weight updates in the current model parallel setting. The weights, gradients, and the number of accumulated gradients are stored locally at each worker and the decision to make an update can be done in a completely decentralized manner. In fact, different parts of a network can be updated with different interval. We used this flexibility to update the word embedding matrix more slowly than other parts of the network for Tree RNN.
> > >
> > > [Open source software]
> > > Our work is neither a simulator nor just a back-of-envelope calculation. We have built the intermediate representation, the multi-threaded runtime and a python frontend for easy model definition. We will open source our project around the time of paper notification.
> > >
> > > [Which system does it make sense?]
> > > The propose AMP training makes sense for a system composed of many computational devices that can act independently. A good example is many FPGAs with high-bandwidth inter-connect. On the other hand, we cannot expect a single GPU to benefit from the proposed idea because although a single GPU has many cores it is challenging to make them act independently.
> > >
> > > [1] Goyal et al. (2017) Accurate, Large Minibatch SGD: Training ImageNet in 1 Hour
> > > [2] Bottou et al. (2017) Optimization Methods for Large-Scale Machine Learning

---

### Official Review · AnonReviewer2 · 2017-11-27
**New approach to asynchrony**

**Rating:** 6
**Confidence:** 5

**Review:**

This paper proposes new direction for asynchronous training. While many synchronous and asynchronous approaches for data parallelism have been proposed and implemented in the past, the space of asynchronous model parallelism hasn't really been explored before. This paper discusses an implementation of this approach and compares the results on dynamic neural networks as compared to existing parallel approaches.

Pros:
- Paper seems to cover and contrast well with the existing approaches and is able to clarify where it differs from existing papers.
- The new approach seems to show positive results on certain dynamic neural network problems.

Cons:
- Data parallelism is a very commonly used technique for scaling. While the paper mentions support for it, the results are only showed on a toy problem, and it is unclear that it will work well for real problems. It will be great to see more results that use multiple replicas.
- As the authors mention the messages also encapsulate meta-data or "state" as the authors refer to it. This does seem to make their compiler more complex. This doesn't seem to be a requirement for their design and proposal, and it will be good to see explorations to improve on this in the future.
- Comparisons with Dynet (somewhat hidden away) that offers auto-batching in a dynamic mode aren't very positive.

Questions:
- It appears that only a single copy of the parameters is kept, thus it is possible that some of the gradients may be computed with newer values than what the forward computation used. Is this true? Does this cause convergence issues?

Overall it seems like a valuable area for exploration, especially given the growing interest in dynamic neural networks.

[Update] Lowered rating based on other feedback and revisiting empirical results. The ideas are still interesting, but the empirical results are less convincing.

---

> ### Author Response · Authors · 2017-12-06
> **reply**
>
> Thank you for your careful review. Here is our response to your questions:
>
>  1. Data parallelism (replicas) can be easily deployed in any model (see Section 5).
>
>  2. State (or metadata) is necessary in the proposed asynchronous setup. For example, take an operation that receives forward propagations from multiple parents (e.g., add or concat); messages from different parents and different training instances may arrive in any order due to asynchrony and we need to guarantee that the messages are correctly grouped by the instance id or the loop counter (if the operation lies inside a loop) stored in the state. The same is true for the backward phase of an operation that receives messages from multiple children. A global scheduler could be used instead of state in a more centralized system but that would require more communication. The focus of our paper is to explore a decentralized system that doesn't require a scheduler.
>
>  3. The reviewer is absolutely correct about the possible impact of the staleness of the gradients. We studied this empirically in the paper (see Fig 5), and we found that the staleness can be controlled by either reducing max_active_keys (the maximum number of examples that the system can process at any given moment) or increasing the min_update_interval (number of gradients to accumulate before applying updates). When the staleness was reasonably small, the convergence was minimally affected (e.g., max_active_keys=8 for 8 replicas in Figure 5). Moreover, in our multi-core CPU implementation we found that in most cases there is no benefit in increasing max_active_keys larger than the number of cores (16 in this example).

---

### Author Response · Authors · 2018-01-05
**New version**

Dear reviewers and area chairs,

Thank you for all your comments. We have updated the submission with the following changes:
 * The quantitative claims about the hypothetical 1 TFLOPS device have been moved out of the main text and now appears in Appendix C (following AnonReviewer1's suggestion)
 * We have added Fig. 5 (a) to clarify the difference between pure pipeline parallelism and asynchronous model parallelism and to highlight the fundamental trade-off we are addressing in this paper.
 * We have also included preliminary comparison against DyNet in Section 6. Hopefuly this will serve as an additional data point supporting our asynchronous training working with a network with dynamic routing decisions.  More details are given in Appendix B.5.

---

### Decision · Program_Chairs · 2018-01-29
**ICLR 2018 Conference Acceptance Decision**

**Decision:**

Reject

**Comment:**

The authors propose a system for asynchronous, model-parallel training, suitable for dynamic neural networks.  To summarize the reviewers:

PROS:
1. Paper contrasts well with existing work.
2. Positive results on dynamic neural network problems.
3. Well written and clear

CONS:
1. Some concern about extrapolations/estimates to hardware other than that on CPU.
2. Comparisons with Dynet seem to suggest auto-batching results in a dynamic mode aren't very positive.

For 1) the AC notes the author's objections to reviewer 1's views on the value of estimation/extrapolation to non-CPU hardware.  However, reviewer 3 voiced  a similar concern and  both still feel that there is more to be done to be convincing in the experiments.